

# A novel peroxisome-related gene signature predicts clinical prognosis and is associated with immune microenvironment in low-grade glioma

Dandan Gao[1,*], Qiangyi Zhou[2,*], Dianqi Hou[2], Xiaoqing Zhang[1], Yiqin Ge[3], Qingwei Zhu[2], Jian Yin[2], Xiangqian Qi[2], Yaohua Liu[2], Meiqing Lou[2], Li Zhou[4] and Yunke Bi[2]

[1] Oncology and Hematology, Shanghai University of Medicine & Health Sciences Affiliated Zhoupu Hospital, Shanghai, China
[2] Neurosurgery, Shanghai General Hospital, Shanghai, China
[3] Department of Neurosurgery, Jiading Branch of Shanghai General Hospital, Shanghai Jiao Tong University School of Medicine, Shanghai, China
[4] Department of Oncology, Shanghai Songjiang District Central Hospital, Shanghai, China
[*] These authors contributed equally to this work.

Corresponding authors
Li Zhou, lzhou_sh@hotmail.com
Yunke Bi, doctorbyk@163.com

## ABSTRACT

Low-grade glioma (LGG), a common primary tumor, mainly originates from astrocytes and oligodendrocytes. Increasing evidence has shown that peroxisomes function in the regulation of tumorigenesis and development of cancer. However, the prognostic value of peroxisome-related genes (PRGs) in LGG has not been reported. Therefore, it is necessary to construct a prognostic risk model for LGG patients based on the expression profiles of peroxisome-related genes. Our study mainly concentrated on developing a peroxisome-related gene signature for overall survival (OS) prediction in LGG patients. First, according to these peroxisome-related genes, all LGG patients from The Cancer Genome Atlas (TCGA) database could be divided into two subtypes. Univariate Cox regression analysis was used to find prognostic peroxisome-related genes in TCGA_LGG dataset, and least absolute shrinkage and selection operator Cox regression analysis was employed to establish a 14-gene signature. The risk score based on the signature was positively associated with unfavorable prognosis. Then, multivariate Cox regression incorporating additional clinical characteristics showed that the 14-gene signature was an independent predictor of LGG. Time-dependent ROC curves revealed good performance of this prognostic signature in LGG patients. The performance about predicting OS of LGG was validated using the GSE107850 dataset derived from the Gene Expression Omnibus (GEO) database. Furethermore, we constructed a nomogram model based on the gene signature and age, which showed a better prognostic power. Gene ontology (GO) and Kyoto Encyclopedia of Genes and Genomes (KEGG) analyses showed that neuroactive ligand-receptor interaction and phagosome were enriched and that the immune status was decreased in the high-risk group. Finally, cell counting kit-8 (CCK8) were used to detect cell proliferation of U251 and A172 cells. Inhibition of ATAD1 (ATPase family AAA domain-containing 1) and ACBD5 (Acyl-CoA binding-domain-containing-5) expression led to significant inhibition of U251 and A172 cell proliferation. Flow cytometry detection showed that ATAD1 and ACBD5 could induce apoptosis of U251 and A172 cells. Therefore, through

bioinformatics methods and cell experiments, our study developed a new peroxisome-related gene signature that migh t help improve personalized OS prediction in LGG patients.

# INTRODUCTION

Glioma derived from brain glial cells is the most common primary tumor of the central nervous system (CNS) and has a high degree of malignancy (*Reni et al., 2017*). The World Health Organization (WHO, 2016) divides glioma into grades I–IV, of which grades I and II are treated as LGG, accounting for 40%–50% of tumors in patients under the age of 18. LGG tumors are characterized by a slow growth rate and may even show periods of growth arrest. LGG is generally treated by comprehensive therapy, including surgery, radiotherapy, and chemotherapy, but none of these methods are capable of curing LGG. Existing therapy methods can only improve patient symptoms, but often result in drug resistance and tumor recurrence, with over half of LGG patients developing high-grade LGG that is difficult to treat. LGG prognosis is also affected by patient age, nerve damage, excision range, gene phenotype, and other factors. Considering the limitations of current LGG treatments, novel therapeutic targets are needed to increase the clinical outcome of LGG, and reliable new prognostic gene signatures are needed to make targeted LGG therapies.

Peroxisomes are single membrane-enclosed organelles that function in the metabolic process of reactive oxygen species, bile acids, ether phospholipids, branched-chain, and long chain fatty acids (*Schrader & Fahimi, 2006*). Abnormal metabolism is a hallmark of most cancers (*Benedetti et al., 2010*; *Laurenti et al., 2011*; *Pavlova & Thompson, 2016*). In recent years, many studies have demonstrated that peroxisomes play a positive role in cancer (*Dahabieh et al., 2018*). Enzymes that participate in peroxisomal lipid processing are increased in various types of cancers, including prostate cancer (*Zha et al., 2005*), colorectal cancer (*Gupta et al., 2001*), breast cancer (*Fenner & Elstner, 2005*), liver cancer (*Peters, Cheung & Gonzalez, 2005*), ovarian cancer (*Vignati et al., 2006*), glioma (*Bruns et al., 2019*), glioblastoma (*Hua et al., 2020*; *Laurenti et al., 2011*), and bladder cancer (*Inamoto, Shah & Kamat, 2009*; *Mansure, Nassim & Kassouf, 2009*). In addition, employing *in vivo* mouse models, controlling the expression of genes participating in peroxisome degradation and/or chemically restraining peroxisomal lipid processing has shown inhibited tumor growth across diverse cancer types.

This study constructed a new prognostic signature based on peroxisome-related genes in LGG cohorts through univariate Cox and LASSO Cox regression analyses. This signature was then validated as a robust, independent predictor for risk stratification in LGG patients. A nomogram was also constructed combining gene signature and patient age that had better prognostic value in LGG patients. Finally, CCK8 and flow cytometry were used to explore the biological function of *ATAD1* and *ACBD5* in glioma cells.

## MATERIALS AND METHODS

### Data collection of known peroxisome-associated genes (PRGs)

A total of 113 PRGs were collected from the human liver peroxisomes, Peroxisome DB 2.0 database, and KEGG database.

### Datasets

The transcriptional gene expression profiles of 511 LGG patients from the TCGA_LGG cohort were produced using the Illumina HiSeq RNA-Seq platform and relevant clinical features such as patient age, gender, overall survival (OS) time, and survival status were obtained from the UCSC Xena database (https://xenabrowser.net/datapages/). The gene expression data and corresponding clinical information of the validation dataset were downloaded from the GEO database (GSE107850). The keywords "LGG or Low-grade glioma and gene expression and survival" were used for searching the NCBI GEO database. The eligibility criteria for selecting the suitable dataset were clinical outcome with survival time and mRNA profiles.

### Consensus clustering

To analyze LGG molecular subtypes, the "Consensus Clustering Plus" functional module from Sangerbox Tools (http://sangerbox.com/) was used to split LGG patients into different subtypes. The parameters were: distance –(1-Pearson correlation), 80% sample resampling, and 80% gene resampling.

### Construction and evaluation of peroxisome-related gene signature

A univariate Cox proportional hazards regression analysis was conducted using the "survival" and "survminer" R package and the results revealed the genes that were significantly related to overall survival (OS) in the training cohort. Hazard ratios (HRs) and 95% confident intervals (95% CIs) were also calculated. HR >1, $p < 0.05$ were treated as a positive relationship with event hazards and a negative relationship with survival time.

LASSO (least absolute shrinkage and selection operator) penalized Cox regression was used to establish an optimal risk signature from survival-related genes using the "glmnet" R package. The risk score for every patient in both cohorts was calculated by taking the sum of the LASSO regression coefficient for every signature gene multiplied with its relevant expression value. Patients were then divided into high- and low-risk groups based on the median risk scores in each cohort. A principal component analysis (PCA) was performed, using the "ggplot2" R package, to study the distribution of genes in different groups based on the expression level of genes in this model.

To assess the prediction efficiency of this signature, time-dependent ROC (receiver-operating characteristic) and AUC (calculated the area under the curve) analyses were performed using R packages. The "timeROC" R package was used to conduct 1-year, 3-year, and 5-year ROC analyses.

## Validation of the gene signature's prognostic value

An LGG cohort from the GEO database (GSE107850) was used to validate the prognostic value of the gene signature identified in this study with an external dataset. The expression of each peroxisome-related gene was normalized using the "scale" function and the risk score was then calculated using the same formula that was used for the TCGA cohort. Based on the best risk score, the patients in the GSE107850 cohort were also divided into low- or high-risk subgroups, and these groups were then compared to validate the gene model.

## Independent prognostic value of the gene signature

Patient age was extracted from the clinical information of LGG patients in the TCGA dataset. This variable was then analyzed in combination with the risk score in the regression model. Univariate and multivariable Cox regression analyses were also performed.

## Bioinformatics analysis of the differentially expressed genes (DEGs) between the low- and high-risk groups

LGG patients from the TCGA cohort were divided into two subgroups based on the best risk score. The DEGs between the low- and high-risk groups were filtered based on specific criteria ($|log2FC| \geq 1$ and FDR <0.05) using the "Bioconductor Limma" R package. GO and KEGG analyses were performed for these DEGS by applying the Sangerbox Tools (http://sangerbox.com/). A PPI (Protein-protein interaction) network analysis was performed using the STRING (Search Tool for the Retrieval of Interacting Genes/Proteins) protein interaction database (https://cn.string-db.org/).

## Comprehensive analysis of tumor microenvironment

CIBERSORTx (https://cibersort.stanford.edu/) was used to calculate the abundance of 22 immune cell types for clarifying the association between peroxisome-related genes and immune infiltration in the TCGA_LGG dataset.

## Construction and assessment of the nomogram

The risk scores and clinical characteristics that were found to be significant in the univariate Cox analysis were selected to establish a nomogram using the "survival" and "rms" R packages. The concordance index (C-index) was used to assess the performance of this model. Calibration plots were made to assess the concordance between actual and predicted survival.

## Cell culture and transfection

U251 and U87 cells are two common human glioma cell lines, with some differences in morphological, biological, and molecular characteristics. Many studies have used U251 and U87 cell lines to study the molecular mechanisms of specific genes. For example, the gene expression of ATAD1 and ACBD5 are both upregulated in these two cells. For this study, two glioma cell lines (U251 and A172) were purchased from the cell bank of the Chinese Academy of Sciences (Shanghai, China). These two cells were cultured in DMEM medium with 10% FBS, 100 U/mL penicillin, and 0.1 mg/mL streptomycin added at 37 °C

under a 5% CO2 environment. U251 and A172 cells under logarithmic phase status were digested and then planted into two 6-well plates.

ATAD1 and ACBD5 siRNA was transfected into U251 and A172 cells using Lipofectamine 2000, according to the instructions. A scrambled RNA was used as the negative control. The ATAD1_si and ACBD5_si sequences were:
ATAD1-si1:GAAGCAAAUUGGAGUGAAAtt;
ATAD1-si2:GAAUGAAGUUGUUGGUUUAtt;
ATAD1-si3:CAUGUUACUUGGAGUGAUAtt;
ACBD5-si1:GCAUUCACCAAGAUAUAAAtt;
ACBD5-si2:CCGUUAAUGGUAAAGCUGAAAtt;
ACBD5-si3:GCACAGUGGUUGGUGUAUUUAtt

### Fluorescence quantitative PCR

A TRNzol universal RNA extraction kit (Tiangen, Beijing, China) was used to extract total RNA of ATAD1-si3/siNC transfected U251 cells and ATAD1-si3/siNC transfected A172 cells. Total RNA was then reverse transcribed to cDNA using a PrimeScript RT reagent kit with gDNA Eraser (Takara, Tokyo, Japan). PCR was then performed in a TB Green Premix Ex Taq II (Tli RNaseH Plus; Takara, Tokyo, Japan). The $2^{-\Delta\Delta Ct}$ method was then used to analyze the gene expression data. The qPCR primers for *ATAD1* and *ACBD5* were:
ATAD1_upstream: 5′-GCTACCAATCGTCCTCAGGA-3′;
ATAD1_downstream: 5′-TTCCTGGGCAACTTCTAGCA-3′;
ACBD5_upstream: 5′-GCCTTGTCCGGCAATACCAA-3′;
ACBD5_downstream: 5′-CGGCAAACTCTGGATCACCT-3′;
$\beta$-Actin-F: 5′-CATCCGCAAAGACCTGTACG-3′
$\beta$-Actin-R: 5′-CCTGCTTGCTGATCCACATC-3′

### CCK8 assay

CCK8 (Beyotime Biotechnology, Jiangsu, China) was used to detect cell viability according to the manufacturer's instructions: 10,000 U251 or A172 cells were planted into each well of a 96-well plate. Then, 10 µL CCK8 solution was added and incubated for 2 h at each of the following time points: 24 h, 48 h, 72 h, 96 h, and 120 h after transfection. Cell viability was calculated by measuring OD 450.

### Apoptosis detection

A Calcein-AM/PI kit (DOJINDO) was used to detect cell apoptosis. After transfection for 48 h, U251 or A172 cells were collected and washed with PBS buffer. These tumor cells were then centrifuged and the supernatant was carefully removed. Annexin V binding buffer was added to resuspend cells to $5 \times 10^6$/mL. Then, the 5 µL Annexin V/FITC mix with cell suspension (100 µL) was incubated for 5 min, and 10 µL PI dye and 400 µL PBS was added before flow cytometry.

### Statistical interpretation

To analyze differences in the OS of patients between subgroups, a survival analysis was performed using the Kaplan–Meier method with a two-sided log-rank test. Univariate and

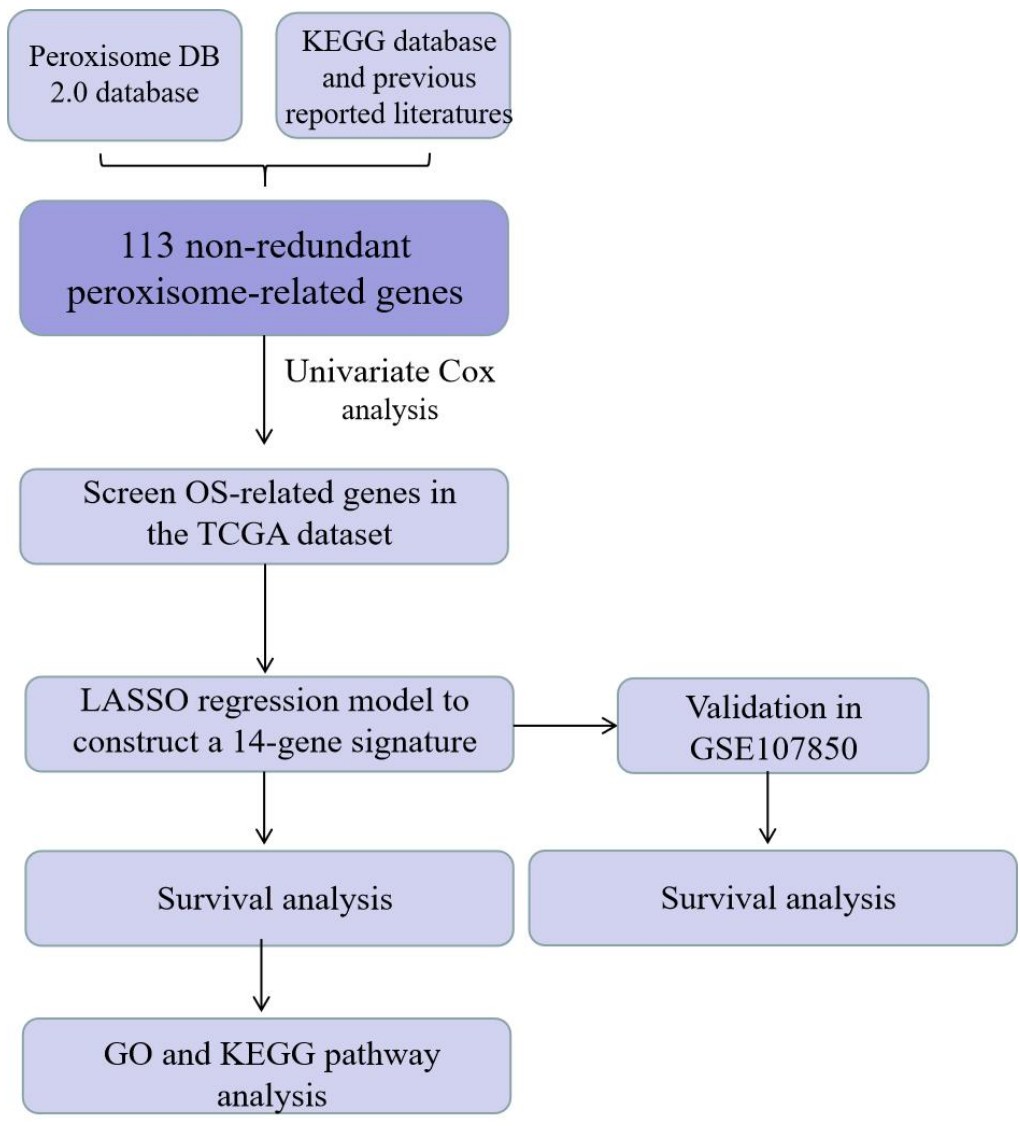

**Figure 1** The data analysis workflow diagram.

multivariate Cox regression models were used to explore the independent prognostic value of the risk model. A Mann–Whitney test was used to compare immune cell infiltration between the two groups. All statistical analyses were performed using the R software (4.1.1; *R Core Team, 2022*), and *P* < 0.05 was considered statistically significant. The overall flow diagram is shown in Fig. 1.

# RESULTS

## Clinical characteristics of LGG patients

Data from 706 LGG patients from the TCGA-LGG cohort ($n = 511$) and the GSE107850 cohort ($n = 195$) were used in this study. Detailed clinical information of the LGG patients in these two cohorts is shown in Table 1.

**Table 1  Characteristics of patients with LGG in the TCGA and GSE107850 datasets.**

| Characteristic | TCGA cohort | GSE107850 |
|---|---|---|
| Gender | | |
| male | 283 (55.38%) | 110 (56.41%) |
| female | 228 (44.62%) | 85 (43.59%) |
| Age | | |
| <60 | 442 (86.50) | 178 (91.28%) |
| >=60 | 69 (13.50%) | 17 (8.72%) |

## Tumor classification according to the peroxisome-related genes

A consensus cluster analysis was performed with 511 LGG patients from the TCGA dataset to study the relationship between the expression of 113 peroxisome-related genes and LGG subtypes. Increasing the k value (clustering variable) from two to 10 showed that when $k = 2$, 511 LGG patients could be perfectly divided into two clusters based on the 113 peroxisome-related genes (Fig. 2A). Differences in OS time were also compared between these two groups, and the results showed that the LGG patients in Cluster 2 (C2) had a lower survival rate than those in Cluster 1 (C1; Fig. 2A).

## Construction of a prognostic gene signature based on the LGG_TCGA cohort

Samples from 511 LGG patients were matched with relevant patients who had complete survival clinical information. A univariate Cox regression analysis was used to screen genes related to survival. The 65 genes that met the criteria of $P < 0.0$ were reserved for further analysis. Among these genes, 33 were found to be associated with higher risk (HRs >1), while the other 32 genes were associated with lower risk (HRs <1). A multivariate Cox analysis was then performed to evaluate whether the expressions of these 65 genes could be used as an independent prognostic factor in LGG. The results showed that 61 genes could be independent prognostic factors. By performing a LASSO Cox regression analysis, a 14-gene signature, including ACBD5, ACSL1, ACSL5, ATAD1, CROT, DHRS4, GRHPR, IDH1, IDI1, IDI2, NUDT19, NUDT7, PEX16, and PEX7, was established based on the optimum λ value (Figs. 3A–3B). The risk score was calculated as follows: risk score = (−0.519*ACBD5 exp.) + (0.030*ACSL1 exp.) + (0.053* ACSL5 exp.) + (−0.226*ATAD1 exp.) + (0.221*CROT exp.) + (−0.022*DHRS4 exp.) + (−0.011*GRHPR exp.) + (0.065*IDH1 exp.) + ( −0.132*IDI1 exp.) + (−0.4071* IDI2 exp.) + (0.074* NUDT19 exp.) + (−0.044* NUDT7 exp.) + (−0.196* PEX16 exp.) + (−0.173* PEX7 exp.).

According to the best cut off score calculated by the risk score formula, the 511 LGG patients were separated into low- and high-risk subgroups (Fig. 3C). PCA results demonstrated that LGG patients with different risks could be divided into two groups even though there was one overlap in the PCA plot (Fig. 3D). There were also more deaths and a shorter survival time in the high-risk patient group than in the low-risk patient group (Fig. 3E). There was a significant difference in OS time between the high-risk and low-risk groups ($P < 0.001$).

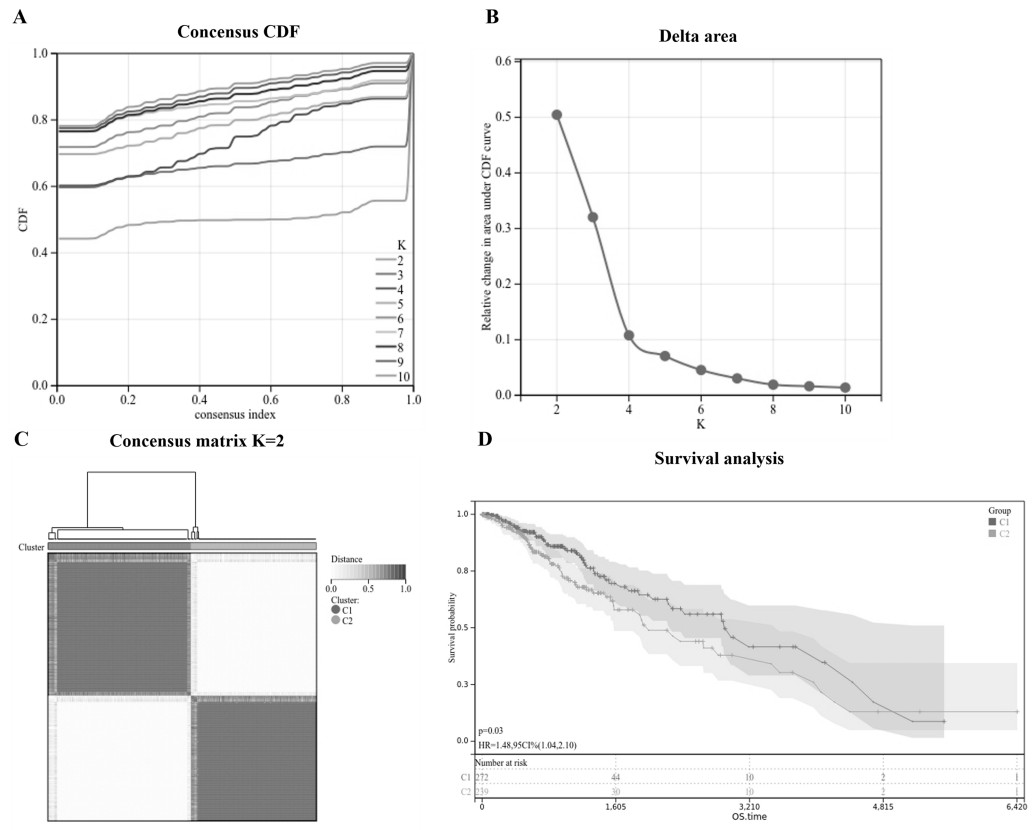

**Figure 2** **Identification of two classifications of LGG in the TCGA_LGG dataset.** (A) Empirical cumulative distribution plot identifying the optimal number of LGG classifications. (B) Relative change in area under the CDF curve was increased. (C) Consensus clustering of LGG patient samples using two classifications. (D) KM plot curves of OS for C1 and C2.

An ROC analysis was used to assess the sensitivity and specificity of this prognostic model. AUC (Area under the curve) was found to be 0.825 for 1-year, 0.831 for 3- year, and 0.752 for 5-year survival (Fig. 3F).

## Validation of the risk gene signature on an independent dataset

A dataset of 195 LGG patients derived from a GEO cohort (GSE107850) was used as the validation dataset (Table 1). These patients were divided into two subgroups based on the best risk score in the GSE107850 dataset, with 49 patients in this cohort in the low-risk subgroup and the other 146 patients in the high-risk subgroup (Fig. 4A). PCA results showed a clearer separation between the two subgroups (Fig. 4B). Patients in the low-risk group (Fig. 4C) had longer survival times and lower death rates than those in the high-risk group. A KM plot analysis also indicated a significant difference in the overall survival rate between the low- and high-risk groups ($P = 0.0048$, Fig. 4D). An ROC analysis of the GEO cohort demonstrated that this model had poor predictive efficacy (AUC = 0.672 for 1-year, 0.546 for 3-year, and 0.547 for 5-year survival; Fig. 4E). This might be due to the small number of LGG patients in the GSE107850 dataset.

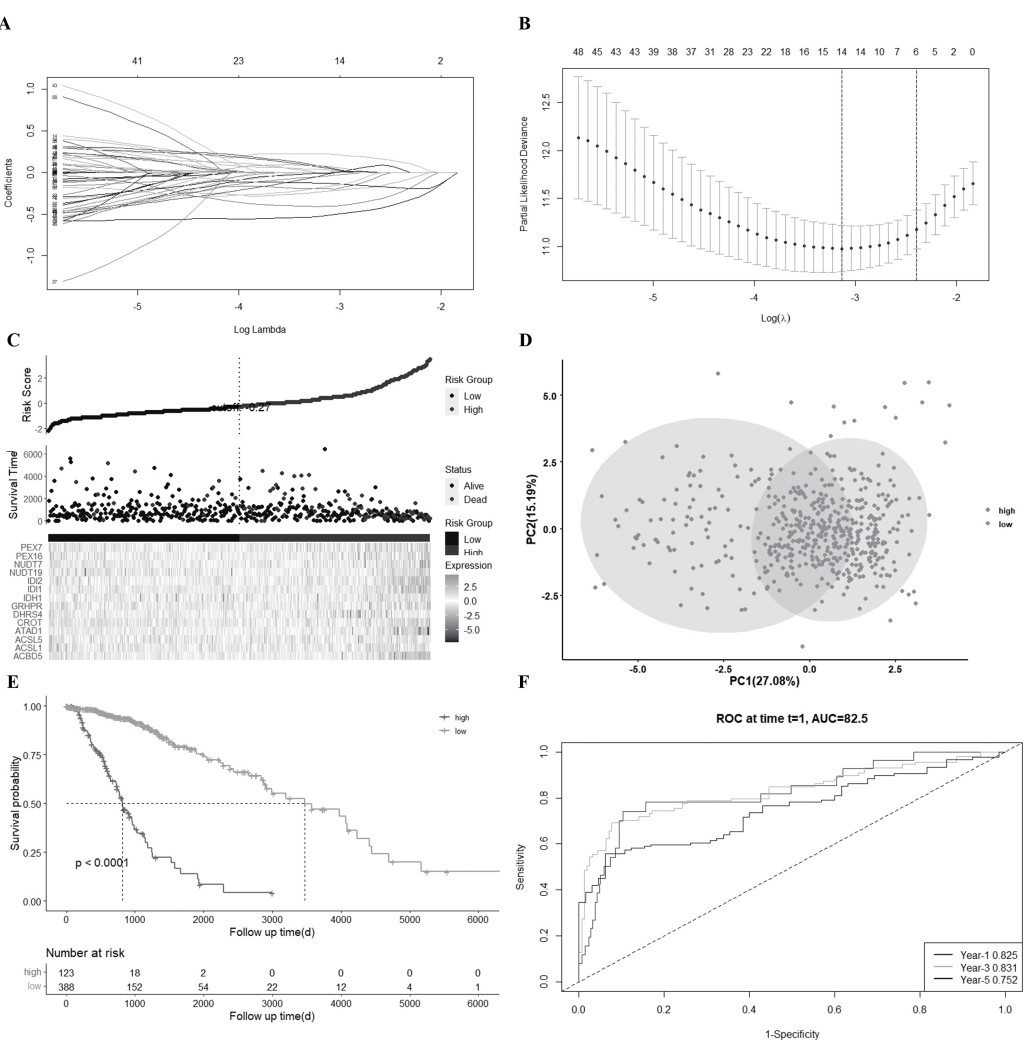

**Figure 3** **Construction of a 14-gene signature in the LGG_TCGA cohort.** (A) LASSO regression of the 14 survival-related genes. (B) Cross-validation in the LASSO regression process. (C) Distribution of LGG patients based on the risk score and survival status of each patient (low-risk population: on the left side of the blue line; high-risk population: on the right side of the red line). (D) PCA plot for LGG patients according to risk score. (E) KM plot for the OS of patients in the high- and low-risk groups. (F) ROC curves showed the predictive efficiency of this risk score.

## Independent prognostic value of the gene signature

Univariate and multivariable Cox regression analyses were used to explore the independent prognostic value of the risk score based on the gene signature model in LGG. The univariate Cox analysis showed that LGG patients with a high risk score had poor survival rates in the TCGA dataset. After adjusting for other confounding factors, the multivariate Cox analysis also showed that the risk score was a prognostic factor for LGG patients. A complex heatmap of clinical characteristics was then constructed for the TCGA dataset (Fig. 5C), which showed that patient age and living status were diversely distributed between the low- and high-risk groups.

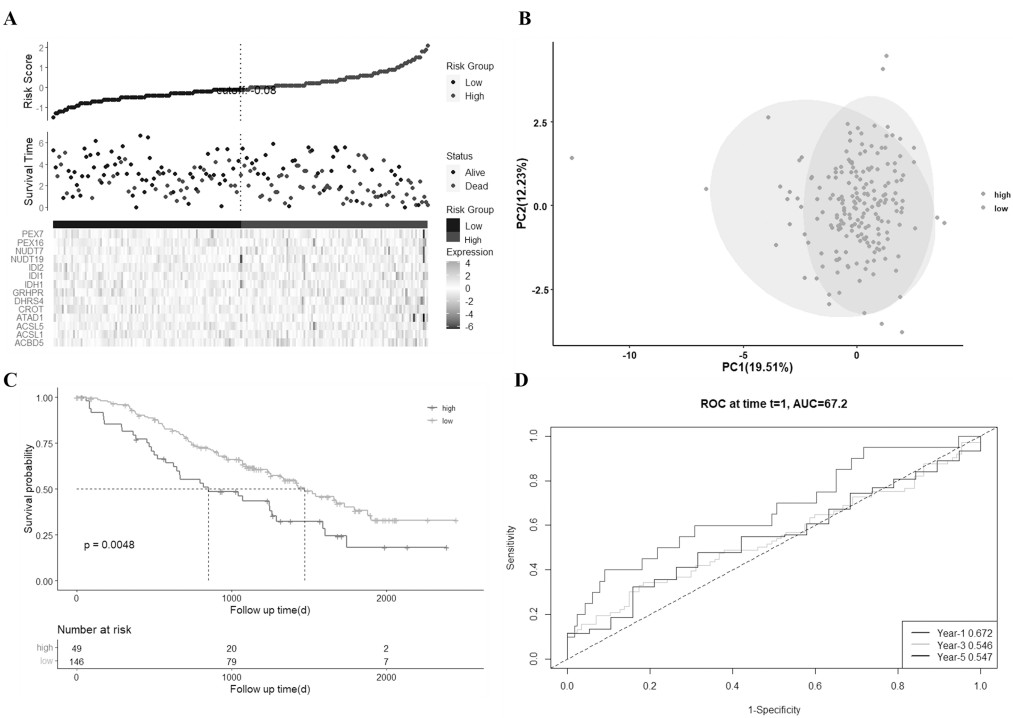

**Figure 4  External validation of the risk model in the GSE107850 dataset.** (A) Distribution of LGG patients in the GSE107850 dataset according to the median risk score in the TCGA_LGG cohort, and the survival status for each patient (low-risk population: on the left side of the blue line; high-risk population: on the right side of the red line). (B) PCA plot for LGG. (C) KM plot for comparing OS between the low- and high-risk groups. (D) Time-dependent ROC curves for LGG patients.

## Construction of a nomogram based on peroxisome-related gene signature

A nomogram was then constructed integrating risk score and the independent clinical risk factor, patient age, in the TCGA_LGG dataset (Fig. 6). Two straight, horizontal straight lines were drawn to show the detailed points for risk score and age, respectively. Then, total points for every patient were calculated by taking the sum of all variate points. The predicted survival probabilities at 1, 2, and 3 years were obtained by drawing a vertical line between the total point line and each prognostic line. The results showed that the predicted and actual survival had a good conformance (Fig. 6). The nomogram indicated that the risk score had a higher weight than patient age. These results indicated that this nomogram may be an optimal model for predicting the prognosis of LGG patients compared to individual risk factors.

## Functional analyses according to the risk model

To further study the differences in the genes' functions and pathways between the subgroups divided by the risk model, the "limma" R package was used to extract the DEGs by applying the criteria FDR <0.05 and |log2FC| ≥ 1. A total of 1,850 DEGs were found between the low- and high-risk groups from the TCGA_LGG database (Table S1). There were 995

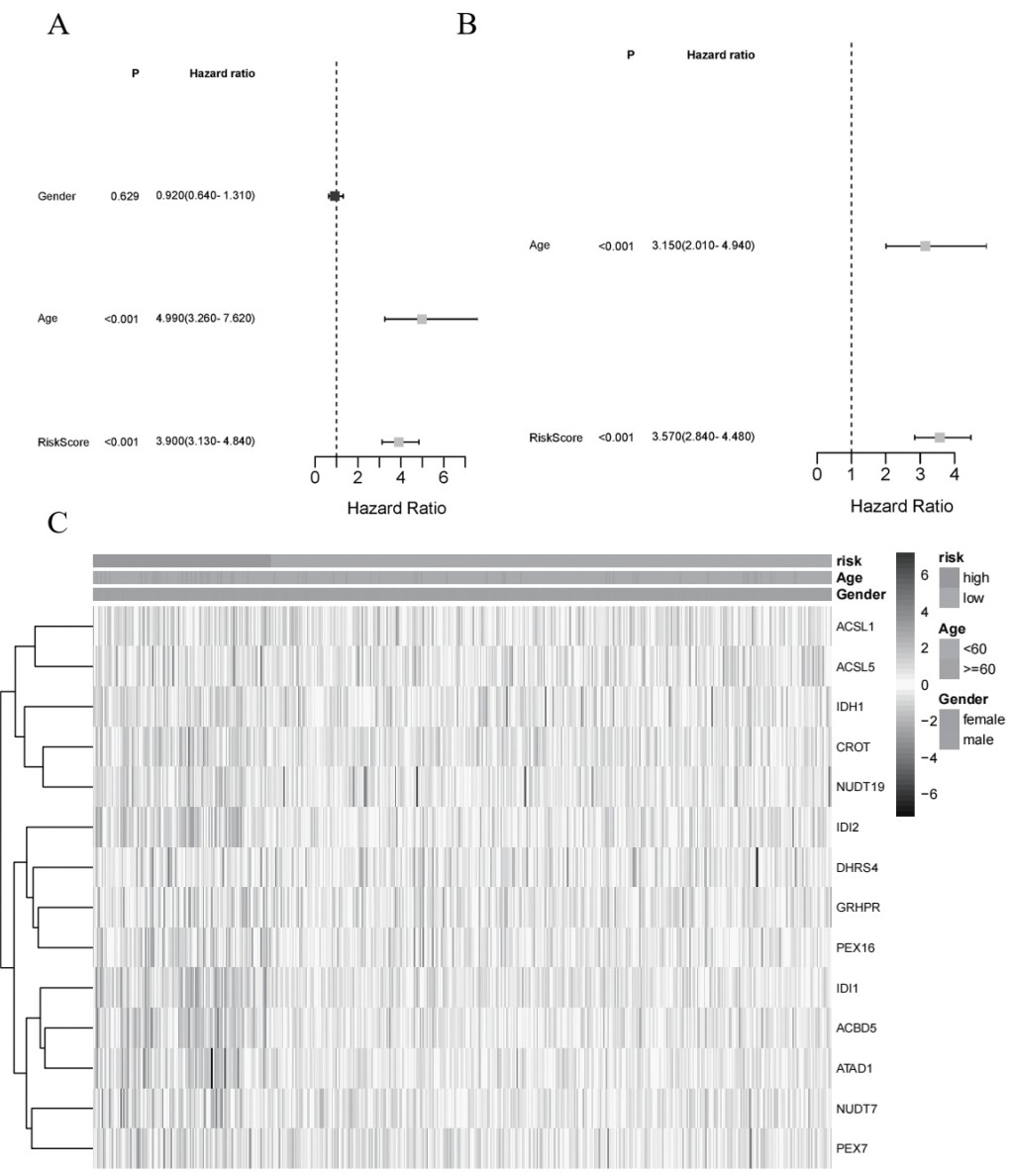

**Figure 5** **Independent prognostic value of the gene signature.** (A) Univariate analysis for the TCGA_LGG dataset (age, gender, risk score). (B) Multivariate Cox analysis for the TCGA_LGG dataset (age, risk score). (C) Heatmap of the expression distribution of 14 DEGs and clinical features between the high-risk and low-risk groups.

upregulated genes and 855 downregulated genes in the high-risk group. A gene ontology (GO) enrichment analysis and Kyoto Encyclopedia of Genes and Genomes (KEGG) pathway analysis were then employed to assess the DEGs. The results demonstrated that these DEGs were mainly associated with the neuroactive ligand–receptor interaction, ECM-receptor interaction, and the phagosome (Fig. 7 and Table S2).

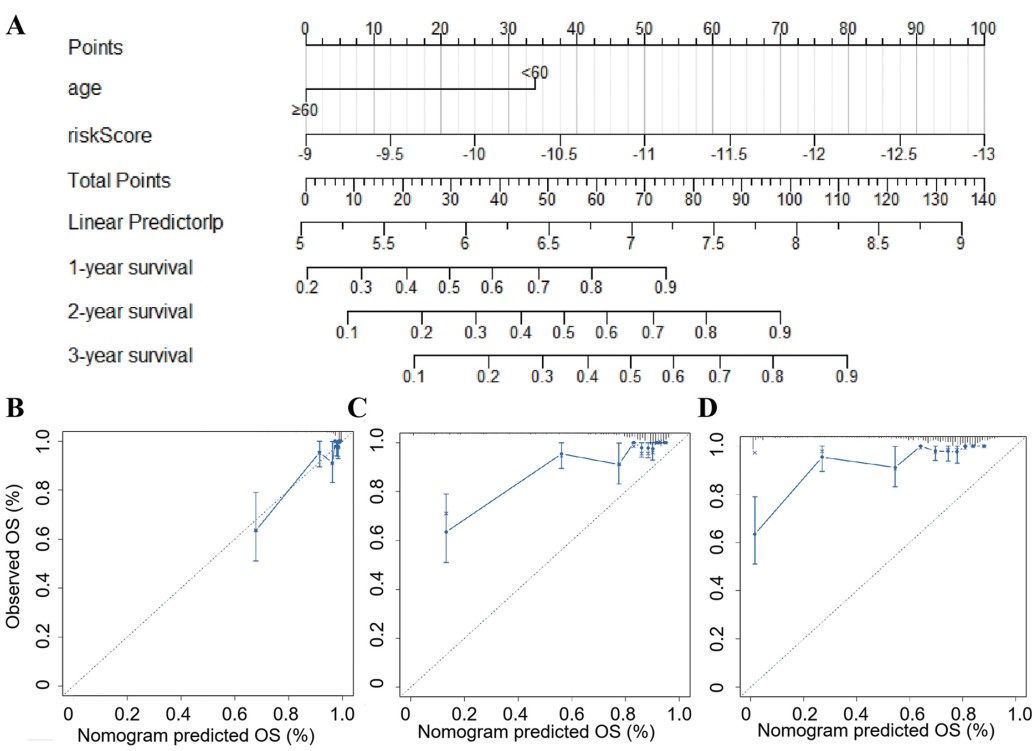

**Figure 6** **Establishment of a new nomogram for OS prediction in LGG patients.** (A) Nomogram combining the 14-gene signature with patient age. (B, C, D) Calibration curve of the nomogram for predicting the probability of OS at 1, 2, and 3 years in the TCGA-LGG dataset.

## Comparison of the immune activity between the high- and low-risk groups

Based on the results of the functional analyses, differences in the enrichment scores of 22 types of immune cells were compared between the low- and high-risk groups in TCGA_LGG cohorts using CIBERSORTx. In the TCGA_LGG dataset, the high-risk subgroup showed lower infiltration levels of immune cells, especially of naïve B cells, plasma cells, naïve CD4 T cells, and monocytes, than the low-risk subgroup (Fig. 7C). Future research should explore the different PRG signatures between astrocytes and oligodendrocytes to determine whether these two cell types have a similar or different LGG-related PRG signature.

## Expression levels of 14 genes in normal and LGG tumor tissues

To explore whether the 14 identified genes could be used as diagnostic biomarkers in LGG, the expression levels of these genes were analyzed in the pooled GTEx (Genotype-Tissue Expression) and TCGA data from 207 normal tissues and 518 tumor tissues. The results showed that compared with normal tissues, the mRNA expression of five genes, ACBD5, ATAD1, DHRS4, GRHPR, and IDH1, was significantly up-regulated in tumor tissues (Figs. 8A–8E). To further study the interactions of these 14 peroxisome-related genes, a PPI analysis was performed and the results are shown in Fig. 8F and Table S3.

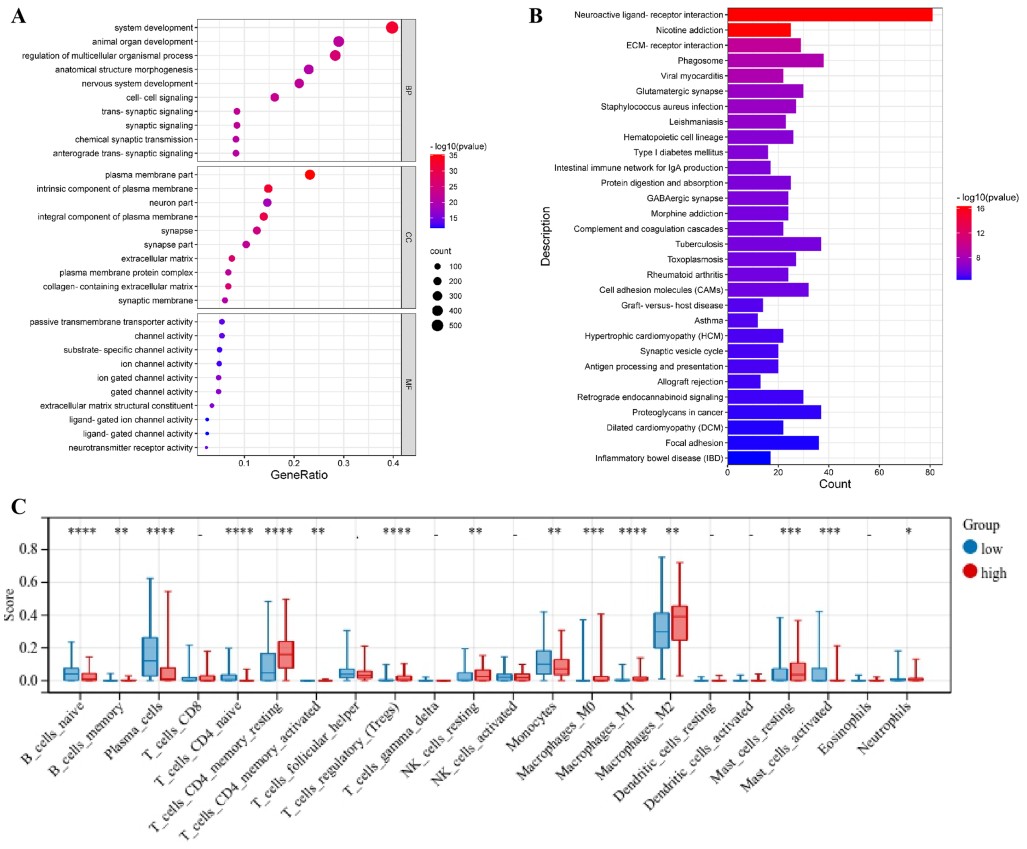

**Figure 7** Functional analysis of DEGs and comparison of the enrichment scores of 22 types of immune cells between the two risk groups in the TCGA_LGG dataset. (A) Bubble graph for GO enrichment. (B) Barplot graph for KEGG pathways. (C) Differences of 22 immune cell subsets in high- and low-risk groups.

## Downregulation of ATAD1 and ACBD5 inhibited U251 and A172 cell proliferation

Because both ATAD1 and ACBD5 were determined to be both diagnostic and prognostic biomarkers in LGG, and these two genes had not been previously reported in LGG, they were selected for cell experiment verification. To investigate the function of ATAD1 or ACBD5, three siRNAs were designed to knock down ATAD1 and ACBD5 in U251 or A172 cells, respectively. The interference efficiencies are quantified in Fig. 9. The results indicated that all three kinds of siRNAs showed the powerful activity in knocking down ATAD1 and ACBD5 expression, respectively. One siRNA, siRNA3 (ATAD1_si3 or ACBD5_si3), was selected for follow-up experiments. The qPCR results, shown in Fig. 10, verified the inhibition of siATAD1 on ATAD1 expression and siACBD5 on ACBD5 expression. Sustained proliferation is the most important characteristic of cancer cells, so the influence of ATAD1 and ACBD5 downregulation on A172 or U251 cell proliferation was assessed using a CCK8 assay. The results, shown in Fig. 11, indicated that after transfection for 48 h

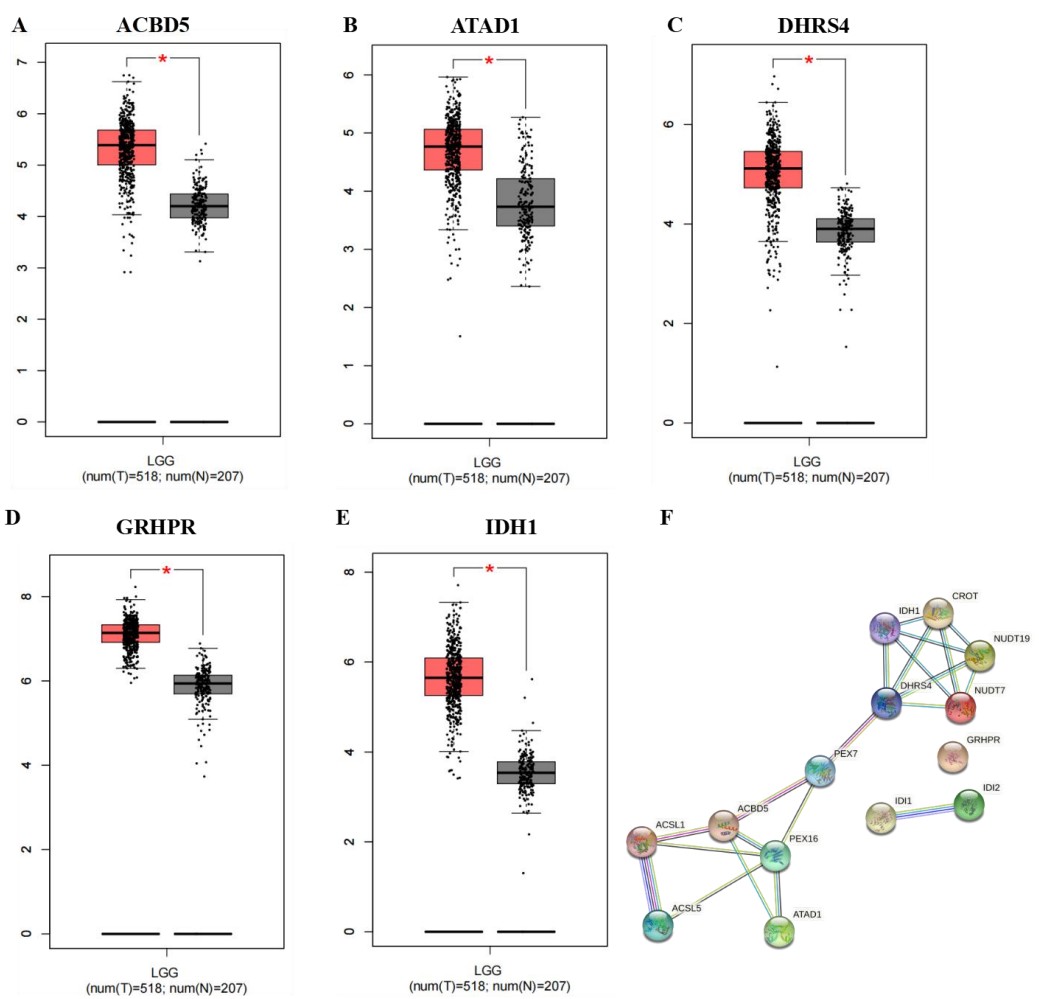

**Figure 8** (A–E) The expression level of five peroxisome-related genes in normal tissue and LGG tumor tissue. (F) Protein–protein interaction network of 14 genes in the STRING database.

and 72 h, the cell viability of ATAD1_si3 and ACBD5_si3 transfected cells was significantly decreased compared to that of siNC transfected cells ($P < 0.05$).

In summary, ATAD1_si3 and ACBD5_si3 transfection inhibited U251 cell proliferation, suggesting that ATAD1 and ACBD5 play an oncogenic role in glioma.

## Downregulation of ATAD1 and ACBD5 induced cell apoptosis

To further explore the inhibition of ATAD1_si3 and ACBD5_si3 transfection on cell proliferation, flow cytometry was performed to analyze apoptosis of A172 or U251 cells transfected with ATAD1_si3 and ACBD5_si3. The results showed that ATAD1_si3 and ACBD5_si3 transfection significantly increased the cell apoptosis percentage in A172 and U251 cells (Figs. 12–13).
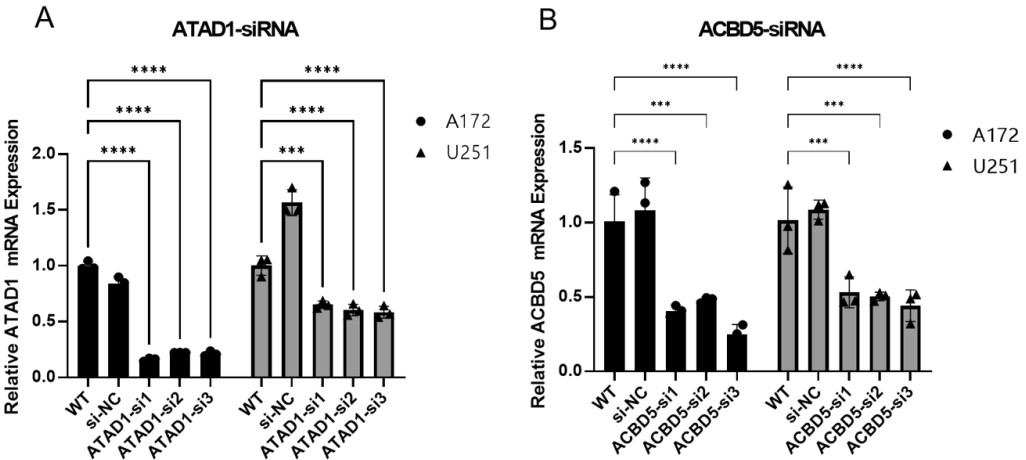

**Figure 9** The qPCR demonstrated that si1, si2, and si3 efficiently inhibited ATAD1 (A) and ACBD5 (B) expression.

## DISCUSSION

The recent development of high-throughput sequencing has significantly improved the understanding of tumor development and progression, and has provided new clues for the accurate diagnosis and prognostic prediction of LGG. However, to date, the number of clinical biomarkers used to predict the survival of LGG patients is very small, restricting the development of early diagnosis and prognosis biomarkers for LGG.

*Özbek et al. (2023)* used bioinformatics methods to identify several diagnostic and prognostic markers (such as CD74, CD86, CDC25A, *etc.*) in LGG. *Liu et al. (2021)* found that a 10-gene signature may be used as a prognostic biomarker for predicting poor prognosis in LGG patients. To further evaluate the prognostic value of these peroxisome-related genes, the present study created a 14-gene risk signature through univariate Cox analysis and LASSO Cox regression analysis, which was also validated in an external GEO dataset. A nomogram was then constructed based on the 14-gene signature that showed a good predictive prognostic ability in LGG patients. A functional analysis demonstrated that the DEGs between the low-risk and high-risk LGG patient groups were related to neuroactive ligand–receptor interaction. This study also compared the infiltrated immune cells in the low- and high-risk groups and found that the high-risk group had decreased levels of infiltrating immune cells compared with the low-risk group.

It is still unclear how PRGs interact in LGG and whether they are associated with the survival time of LGG patients. This study constructed a gene signature featuring 14 peroxisome-related genes (ACBD5, ACSL1, ACSL5, ATAD1, CROT, DHRS4, GRHPR, IDH1, IDI1, IDI2, NUDT19, NUDT7, PEX16, and PEX7) and found that it could predict OS in LGG patients. ACSL1, ACSL5, CROT, IDH1, and NUDT19 were positively related with poor survival, whereas ACBD5, ATAD1, DHRS4, GRHPR, IDI1, IDI2, NUDT7, PEX16, and PEX7 may be protective genes in the TCGA-LGG dataset.

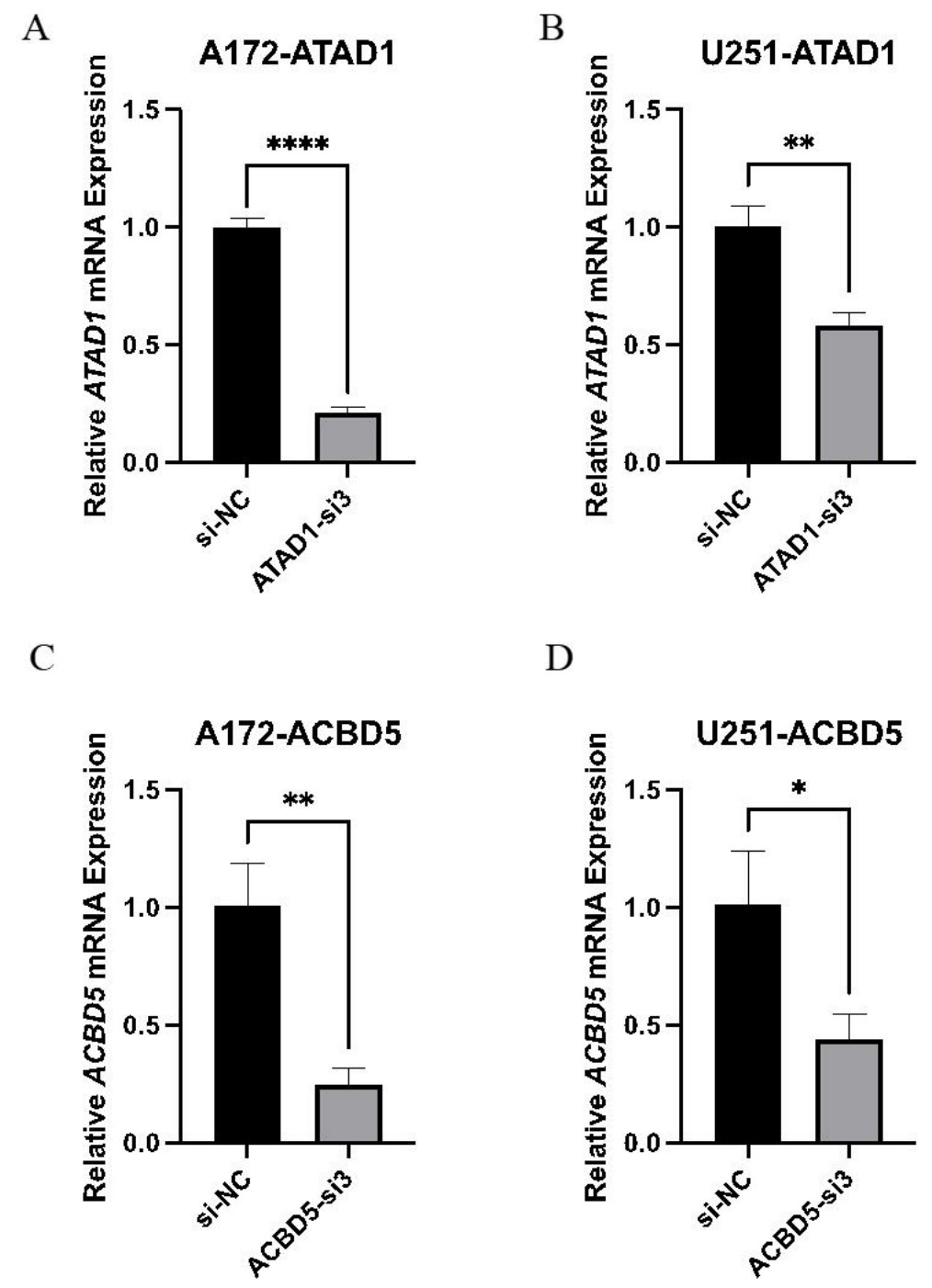

**Figure 10** ATAD1 (A and B) and ACBD5 expression (C and D) was efficiently downregulated by siATAD1 and siACBD5 transfection in A172 and U251 cells.

Fatty acid (FA) metabolism is important for the biogenesis of cellular components and ATP production to maintain cancer cell proliferation. Long-chain fatty acyl-CoA

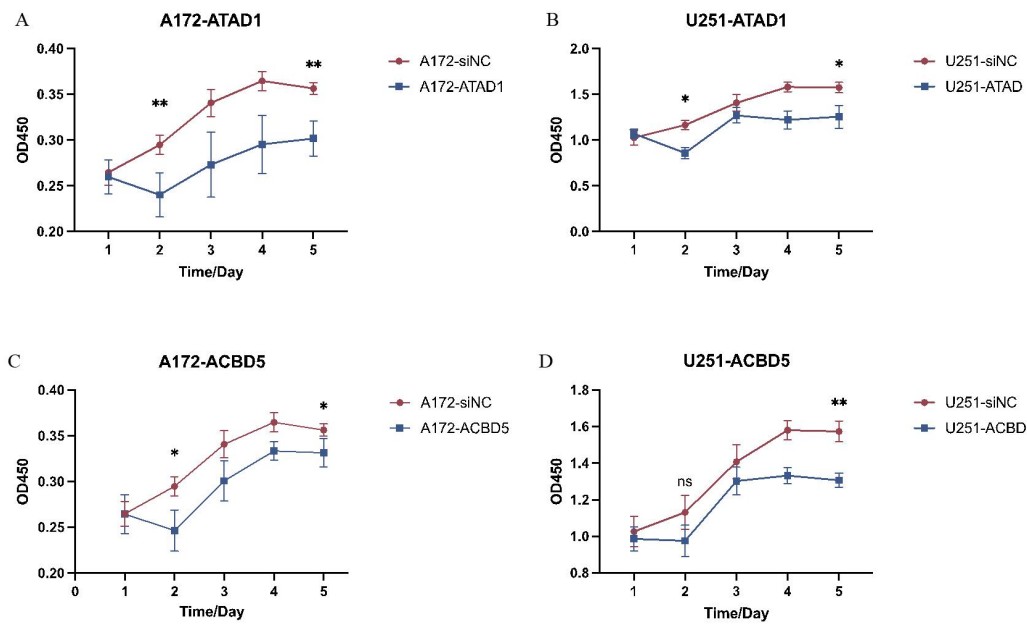

**Figure 11** **Downregulation of ATAD1 (A and B) and ACBD5 (C and D) inhibited cell proliferation in A172 and U251 cells.** A172 and U251 cell proliferation was detected by CCK8 assay.

synthetases (ACSLs) are a group of rate-limiting enzymes in FA metabolism that can convert free-fatty acid to fatty acid-CoA. In 2017, *Wang et al. (2017)* found that oncoprotein hepatitis B X-interacting protein (HBXIP) is able to up-regulate ACSL1 (long-chain fatty acyl-CoA synthetase 1) by activating transcriptional factor Sp1 in breast cancer. In 2019, Reeby et al. showed that ACSL1 can regulate TNF $\alpha$-induced granulocyte-macrophage colony-stimulating factor (GM-CSF) production *via* breast cancer MDA-MB-231 cells (*Thomas et al., 2019*). In 2021, *Zhang et al. (2021)* demonstrated that ACSL1 can promote cancer metastasis by regulating FA metabolism and myristoylation. In the same year, *Ma et al. (2021)* found that ACSL1 can promote prostate cancer progression by elevating lipogenesis and fatty acid beta-oxidation. Therefore, ACSL1 may play an oncogenetic role in different cancer types.

In 2016, *Hartmann et al. (2017)* using a standardized IHC method, found that ACSL5 may serve as an independent prognostic biomarker for early tumor recurrence of sporadic colorectal adenocarcinoma. In 2017, *Yen et al. (2017)* demonstrated that ACSL5 expression is controlled by ER signaling pathways and is a potential novel prognostic biomarker of breast cancer patients. In 2019, *Ma et al. (2019)* using bioinformatics and IHC staining methods, discovered that patients with high ACSL5 expression had a shorter PFS than those with low ACSL5 expression, revealing ACSL5 as a potential prognostic marker in pancreatic cancer patients.

*Lan et al. (2021)* found that NUDT19 regulated by LINC00958 can activate the mTORC1/P70S6K signaling pathway. They also found that both NUDT19 overexpression and mTORC1 activator MYH1485 could reverse the inhibitory effect of LINC00958
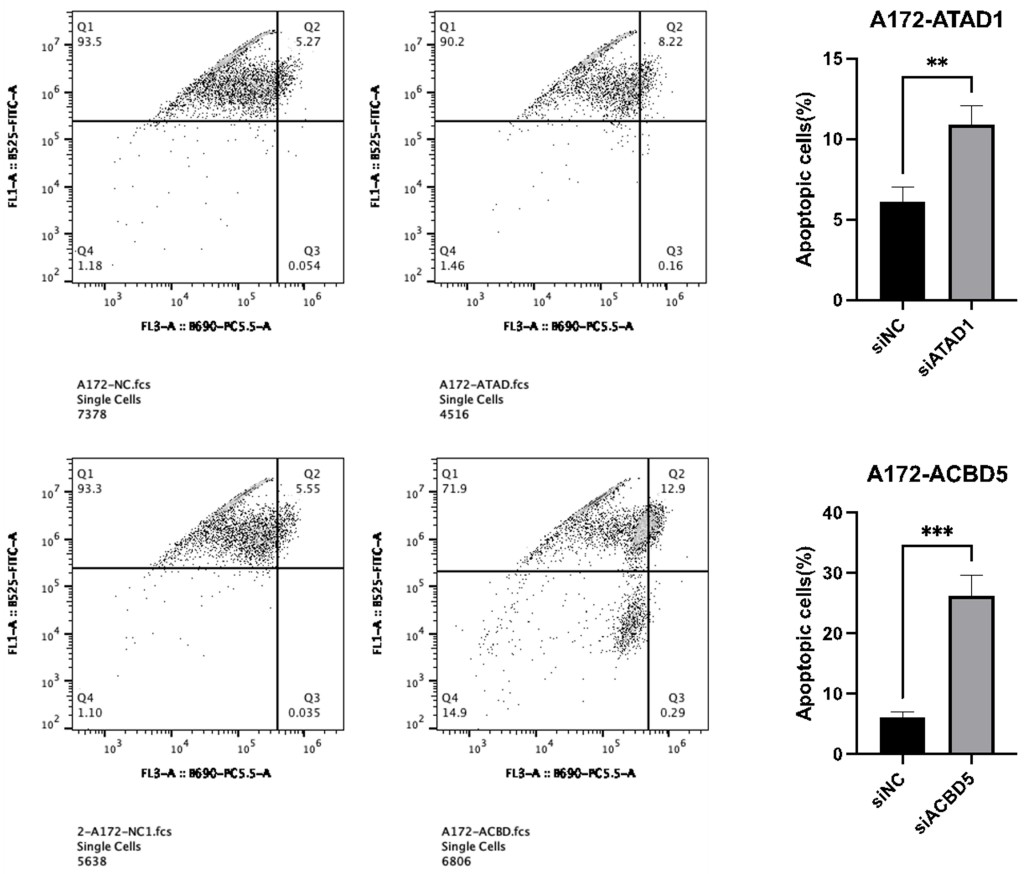

**Figure 12** Downregulation of ATAD1 and ACBD5 induced apoptosis in A172 cells.

silencing on the proliferation, migration, and EMT process of HCC (*Lan et al., 2021*). Notably, in the present study, ACSL1, ACSL5, and NUDT19 seemed to be cancer-promoting genes, as their upregulation was associated with poor survival. However, further study is needed to validate this result.

GRHPR (Glyoxylate reductase/hydroxypyruvate reductase) is an important enzyme in the glyoxylate cycle, and its deficiency can lead to primary hyperoxaluria type 2. *Pan et al. (2013)* demonstrated that patients with negative GRHPR had a significantly shorter survival time than those with positive GRHPR, indicating GRHPR deficiency in noncancerous tissues may be an independent biomarker of worse survival for HCC patients after curative resection. *Song et al. (2020)* found that peroxisomal coenzyme A diphosphatase NUDT7 deletion promotes the development of Kras$^{G12D}$ CRC. These results indicate that GRHPR and NUDT7 are potential protective genes.

The remaining signature genes (ACBD5, ATAD1, CROT, DHRS4, IDH1, IDI1, IDI2, PEX16, and PEX7), which may also be oncogenes or tumor suppressors, have rarely been studied. Further research is required to fully clarify the potential roles and mechanisms of these 14 genes in LGG.

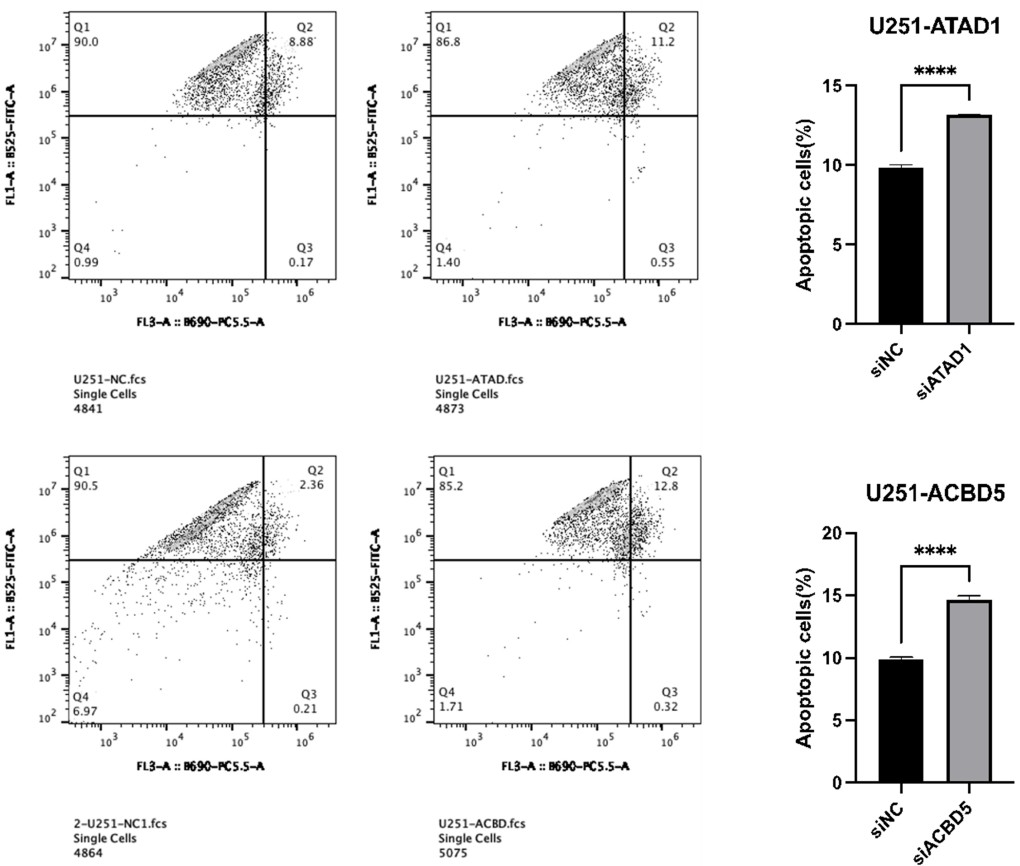

**Figure 13** Downregulation of ATAD1 and ACBD5 induced apoptosis in U251 cells.

This study has some limitations. First, because retrospective data was used, prospective experiment validation of this signature in LGG patients of low-grade glioma or experiments in mouse models is necessary to verify its diagnostic and prognostic potential. Second, the predictive performance of this signature would improve if multi-omics data was appropriately integrated into the analyses. Finally, a large pre-collected cohort is needed to verify the potential prognostic value of the current gene signature.

## CONCLUSION

Although previous studies have reported prognostic PRGs in cancers, this study innovatively established and validated a new prognostic signature containing 14 PRGs that can better predict OS in LGG patients. This 14-gene signature may become a valuable prognostic biomarker for LGG patients, potentially leading to personalized clinical treatment for these patients.

### Funding

Funding for this study came from the National Natural Science Foundation of China (No. 81902529, No. 82072902), the Health Technology Project of Pudong New District Health Committee (PW2020B-2), the Project of National Facility for Translational Medicine (Shanghai) (TMSK-2021-202) and the Scientific Research Funding Project of Shanghai University of Medicine & Health Sciences Affiliated Zhoupu Hospital (ZPXM2019A-01). The funders had no role in study design, data collection and analysis, decision to publish, or preparation of the manuscript.

### Grant Disclosures

The following grant information was disclosed by the authors:
National Natural Science Foundation of China: 81902529, 82072902.
Health Technology Project of Pudong New District Health Committee: PW2020B-2.
Project of National Facility for Translational Medicine (Shanghai): TMSK-2021-202.
Shanghai University of Medicine & Health Sciences Affiliated Zhoupu Hospital: ZPXM-2019A-01.

### Competing Interests

The authors declare there are no competing interests.

### Author Contributions

- Dandan Gao conceived and designed the experiments, prepared figures and/or tables, and approved the final draft.
- Qiangyi Zhou performed the experiments, prepared figures and/or tables, and approved the final draft.
- Dianqi Hou performed the experiments, prepared figures and/or tables, and approved the final draft.
- Xiaoqing Zhang analyzed the data, prepared figures and/or tables, and approved the final draft.
- Yiqin Ge analyzed the data, prepared figures and/or tables, and approved the final draft.
- Qingwei Zhu analyzed the data, prepared figures and/or tables, and approved the final draft.
- Jian Yin performed the experiments, prepared figures and/or tables, and approved the final draft.
- Xiangqian Qi performed the experiments, prepared figures and/or tables, and approved the final draft.
- Yaohua Liu performed the experiments, prepared figures and/or tables, and approved the final draft.
- Meiqing Lou conceived and designed the experiments, authored or reviewed drafts of the article, and approved the final draft.
- Li Zhou conceived and designed the experiments, authored or reviewed drafts of the article, and approved the final draft.

- Yunke Bi conceived and designed the experiments, authored or reviewed drafts of the article, and approved the final draft.

## Data Availability

The data is available at TCGA-LGG and NCBI GEO: GSE107850. The raw measurements are available in the Supplementary File.

## Supplemental Information

Supplemental information for this article can be found online at http://dx.doi.org/10.7717/peerj.16874#supplemental-information.

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
