# Peer review of "A novel peroxisome-related gene signature predicts clinical prognosis and is associated with immune microenvironment in low-grade glioma"

_PeerJ, doi:10.7717/peerj.16874_

## Round 0.1 · original submission · Major Revisions

Authors should revise according to the suggestions of reviewers. The modifications should be marked. A point to point response letter is need.

**Language Note:** The review process has identified that the English language must be improved. PeerJ can provide language editing services - please contact us at [email protected] for pricing (be sure to provide your manuscript number and title). Alternatively, you should make your own arrangements to improve the language quality and provide details in your response letter. – PeerJ Staff

Reviewer 1 ·

Basic reporting

This is an interesting study, where several in vitro and in silico experiments. However, there are several major issues that should be taken into consideration:

Ambiguous and confusing English language was used throughout. The entire manuscript should undergo thorough revision for grammar, syntax and stylistic errors. For example, Line 204: it appears as an unfinished sentence; line 216: “Validation of the risk gene signature in external dataset”, you could change to “...on an independent dataset”; line 269: “...up-expressed...”, replace with “over-expressed” or “up-regulated”; Figure 2 legend: “Relative increase in the area under the CDF curve was increased”, there is redundancy in this sentence.

Figure 1 is shown instead of Figure 4 in the manuscript.

Avoid self-assertive statements: line 292: “In order to determine what factors result in the inhibition of cell proliferation...”, you could rephrase as contribute to the inhibition as they are other unexplored factors that also lead to inhibition.

More literature references relevant to this study should be added such as PMID: 33550903 and PMID: 36945359.

The authors should check more carefully the Instructions for Authors, e.g. the figures in the manuscript are cited both as “Figure” or “Fig.”

Experimental design

The methods are described insufficiently in a way that cannot be replicated. Moreover, the authors do not provide the rationale for choosing the particular methods/databases/software. The authors should mention the URL of the freely accessible databases, the software packages used, as well as the options/option parameters applied. For example, the methods/software (TCGAbiolinks?) used to download, normalize and filter RNA-Seq data from TCGA and GTEx are not mentioned; the ‘map’ accession codes of the KEGG pathways should be provided; the interaction confidence store in STRING should be mentioned; the keywords for searching NCBI GEO database and the eligibility criteria for selecting the GSE107850 dataset.

Validity of the findings

The results are inadequately provided and there are missing output data. Several hasty conclusions, not directly supported by the findings, are inferred as well. For example:

Line 185: “...the expression of 113 peroxisome-related genes...”, the authors should explain whether these are gene co-expression patterns with the same direction of expression or not.

Line 224-225: AUC values ranging from 0.7 to 0.8 are considered acceptable, whereas values below indicate no discriminatory ability. Therefore, the values 0.672, 0.546 and 0.547 are not statistically sound.

In Figure 3C, the low and high risk LGG patients do not form distinct clusters, as there is a major overlap in the PCA plot.

Table 1 is mislabelled, since they mention “...patients with KIRC...” instead of LGG.

Additional comments

The manuscript should be reformatted and undergo extensive revision in order to be publishable.

·

Basic reporting

No comments.

Experimental design

No comments.

Validity of the findings

The validity of findings should be carried out in either the mouse models or patients of low-grade glioma.

Additional comments

The manuscript predicts a 14-gene risk signature related to peroxisome involved in low grade glioma. It can be important finding for prognosis as well as can be useful from therapeutic point of view for LGG patients. However, there are the following concerns.
1) Please explain the rationale for using two cell line U251 and A172. How they are different?
2) Also, what was the rationale for choosing ATAD1 and ACBD5 genes out of 14 genes for cell experiments?
3) Is it possible to differentiate PRG’s signature between astrocytes and oligodendrocytes? Do both cell types have similar or different LGG related PRG’s signature?

---

## Round 0.2 · Minor Revisions

While the authors cite studies related to peroxisome involvement in cancers, there appears to be substantial literature related to the involvement of the peroxisome in glioma but this is not cited. The authors need to include references such as:

Laurenti G, Benedetti E, D'Angelo B, Cristiano L, Cinque B, Raysi S, Alecci M, Cerù MP, Cifone MG, Galzio R, Giordano A, Cimini A. Hypoxia induces peroxisome proliferator-activated receptor α (PPARα) and lipid metabolism peroxisomal enzymes in human glioblastoma cells. J Cell Biochem. 2011 Dec;112(12):3891-901. doi: 10.1002/jcb.23323. PMID: 21866563.

Bruns I, Sauer B, Burger MC, Eriksson J, Hofmann U, Braun Y, Harter PN, Luger AL, Ronellenfitsch MW, Steinbach JP, Rieger J. Disruption of peroxisome proliferator-activated receptor γ coactivator (PGC)-1α reverts key features of the neoplastic phenotype of glioma cells. J Biol Chem. 2019 Mar 1;294(9):3037-3050. doi: 10.1074/jbc.RA118.006993.

Hua, T.N.M., Oh, J., Kim, S. et al. Peroxisome proliferator-activated receptor gamma as a theragnostic target for mesenchymal-type glioblastoma patients. Exp Mol Med 52, 629–642 (2020). https://doi.org/10.1038/s12276-020-0413-1

https://journals.sagepub.com/doi/pdf/10.1177/039463201002300121

And there are likely others. Also, relate how the present research may correlate or contrast to these previous studies."

·

Basic reporting

The manuscript can be accepted.

Experimental design

No comment.

Validity of the findings

No comment.

Additional comments

No comment.

---

## Round 0.3 · Minor Revisions

The authors have improved the manuscript by adding relevant citations. However, the manuscript needs to be carefully edited for typos and English.

---

## Round 0.4 · accepted · Accept

The authors have solved the reviewers' problems perfectly.